# Soluble Extracellular Polymeric Substances Produced by *Parachlorella kessleri* and *Chlorella vulgaris*: Biochemical Characterization and Assessment of Their Cadmium and Lead Sorption Abilities

**DOI:** 10.3390/molecules27217153

**Published:** 2022-10-22

**Authors:** Wioleta Ciempiel, Magdalena Czemierska, Monika Szymańska-Chargot, Artur Zdunek, Dariusz Wiącek, Anna Jarosz-Wilkołazka, Izabela Krzemińska

**Affiliations:** 1Institute of Agrophysics, Polish Academy of Sciences, Doświadczalna 4, 20-290 Lublin, Poland; 2Department of Biochemistry and Biotechnology, Institute of Biological Sciences, Maria Curie-Skłodowska University, Akademicka 19, 20-033 Lublin, Poland

**Keywords:** metal removal, heavy metals, ICP-OES, microalgae

## Abstract

In the present study, the potential of lead and cadmium removal by the extracellular polymeric substances (EPS) produced from *Parachlorella kessleri* and *Chlorella vulgaris* were investigated. Carbohydrates were the dominant components of EPS from both analyzed species. The contents of reducing sugars, uronic acids, and amino acids were higher in EPS synthesized *by C. vulgaris* than in EPS from *P. kessleri*. The analysis of the monosaccharide composition showed the presence of rhamnose, mannose and galactose in the EPS obtained from both species. The ICP-OES (inductively coupled plasma optical emission spectrometry) analyses demonstrated that *C. vulgaris* EPS showed higher sorption capacity in comparison to *P. kessleri* EPS. The sorption capacity of *C. vulgaris* EPS increased with the increase in the amount of metal ions. *P. kessleri* EPS had a maximum sorption capacity in the presence of 100 mg/L of metal ions. The FTIR analysis demonstrated that the carboxyl, hydroxyl, and carbonyl groups of EPS play a key role in the interactions with metal ions. The present study showed *C. vulgaris* EPS can be used as a biosorbent in bioremediation processes due to its biochemical composition, the presence of significant amounts of negatively charged uronic acids, and higher sorption capacity.

## 1. Introduction

Environmental pollution is an effect of progressive industrialization and urbanization. Heavy metals constitute one of the most serious threats among pollutants. Most of the physicochemical methods used for the removal of heavy metals from water environments are characterized by such disadvantages as low efficiency or the generation of sludge [1]. Therefore, the possibility of using microorganisms and their properties in the processes of the removal of harmful substances from polluted environments has been arousing increasing interest. These processes can take place via extracellular biosorption, i.e., binding heavy metal ions by extracellular polysaccharides.

Given their ability to produce extracellular polymeric substances (EPS), microalgae can be used for the biosorption-based purification of aquatic environments. EPS are one of the metabolites synthesized by unicellular algae and secreted by the cell into the surrounding environment. They play an important role in the protection of algal cells against heavy metal toxicity. EPS are complex molecules consisting of carbohydrates, proteins, uronic acids, nucleic acids, and other substances [2].

Lead is a heavy metal and is particularly dangerous to the nervous system, as it penetrates through the blood–brain barrier and competes with Ca and Zn binding sites [3]. Pb accumulated by plants affects their morphology and physiology, e.g., seed germination, seedling growth, cell reproduction, chlorophyll production, and transpiration [4]. Lead, as well as cadmium, has no function in living organisms. Environmental Cd pollution is caused by its industrial application and the use of phosphate fertilizers. Cadmium poisoning may result in multi-organ damage, including the liver, kidneys, adrenal glands, and the hemopoetic system. The toxicity of Cd is related to its affinity to the S-S and –SH groups of proteins and therefore for enzymes and transporter systems. Exposure to this metal leads to the activation of oxidative stress, which results in damage to DNA, protein, and lipids [5].

*Chlorella vulgaris* and *Parachlorella kessleri* are unicellular freshwater green algae representing the family Chlorellaceae from the class Trebouxiophyceae [6,7]. Due to their high growth rate, high lipid productivity, and low tendency to aggregate, these species are considered as a suitable candidate for industrial application [7]. The properties of EPS depend largely on their chemical composition. Lombardi et al. (2005) indicated that the carboxyl groups of EPS are mostly involved in copper complexation [8,9]. Additionally, proteins were found to be important in metal sorption by EPS produced by *C. vulgaris* (Cd and Ag sorption) and *Chlorella pyrenoidosa* (As sorption) [10,11,12]. However, these studies did not determine the content of uronic acids, amino acids, and amino sugars, which also have functional groups that can interact with metal ions. The exopolysaccharide synthesized by *P. kessleri* has not been studied to date in terms of its sorption properties. The knowledge of the sorption properties of EPS from microalgae is important, as the process of heavy metal binding by EPS reduces their toxicity to organisms in the environment. The conventional methods employed in the bioremediation of heavy metals, e.g., chemical precipitation, floatation, or electrochemical methods, often result in secondary pollution [13]. Due to the presence of many functional groups, environmentally friendly microalgal exopolysaccharides are able to bind heavy metals [2].

The aim of this study was to determine the biochemical characteristics of the soluble fraction of extracellular polymeric substances synthesized by unicellular microalgae, *P. kessleri* and *C. vulgaris*. Additionally, the cadmium and lead sorption ability of the obtained EPS was studied. For this purpose, the amount of metal ions bound to the EPS was determined using ICP-OES. Fourier transform infrared spectroscopy (FTIR) was used to identify functional groups involved in the EPS–metal interactions.

## 2. Results and Discussion

### 2.1. EPS Production

The results of the biomass yield as well as the yield and specific productivity of soluble EPS are shown in Table 1. There were no statistically significant differences in the biomass yield between *P. kessleri* and *C. vulgaris*. *P. kessleri* was found to be a more effective EPS producer than *C. vulgaris*. The EPS yield and EPS specific productivity in the case of *P. kessleri* were 16.6% and 21.3%, respectively, higher in comparison to *C. vulgaris*. This allows the formulation of a hypothesis that *P. kessleri* cells are more efficient in EPS synthesis than *C. vulgaris* cells. The results of the present study show that the EPS yield was lower than that reported in the literature [14]. The low productivity determined in the present study may be associated with the separation of the soluble fraction of total EPS. Additionally, the EPS synthesis was analyzed in optimal conditions for cell growth without the application of stress factors, which increase EPS production. The level of EPS production depends on the species and cultivation conditions [2]. The yield of extracellular polymeric substances is also influenced by the extraction and purification methods. In the case of alcohol precipitation, the EPS yield is influenced by the temperature and polarity of alcohol [15]. Since different methods for EPS isolation are used, it is difficult to compare the EPS yields obtained in different studies.

### 2.2. Chemical Composition of EPS

The chemical composition of EPS (Table 1) shows that both studied EPS consist mainly of carbohydrates, which accounted for 57% and 63% of the samples derived from *C. vulgaris* and *P. kessleri,* respectively. The total carbohydrate content was higher in the polymer synthesized by *P. kessleri* (635 µg/mg) compared to EPS produced by *C. vulgaris* (577.3 µg/mg), which indicate that the isolated EPS are extracellular polysaccharides. These results are comparable to those obtained by Capek (2019) for *C. vulgaris* EPS, where the sugar content in EPS was 67% [6]. In this study, the amount of protein in the examined exopolysaccharides was 0.55% and 0.75%, which is lower than the values reported by the literature data [14]. The results showed statistically significant differences in the content of reducing sugars, uronic acids, and amino acids between the EPS of *C. vulgaris* and *P. kessleri*. The hydrolyzed EPS of both analyzed species consisted mainly of reducing sugars and uronic acids. The analysis showed higher content of reducing sugars, uronic acids and amino acids in the EPS synthesized by *C. vulgaris.* The same levels of amino sugars were detected in both samples (Table 1).

The content of uronic acids in this study are in agreement with the results obtained for *C. vulgaris* EPS by Ogawa (1999), who reported the level of 14% of glucuronic acid [16]. In turn, El-Naggar (2020) found that the content of uronic acid in polysaccharides extracted from *C. vulgaris* was 17% [17]. Other investigations have demonstrated that the content of uronic acids in microalgal EPS may vary. Differences in the uronic acid content in EPS, ranging from 6.0% to 14% were observed for microalgal strains from the family *Dictyosphaerium* [18]. It has been reported that the content of uronic acids is one of crucial factors for the ability of EPS to bind metal ions [8]. As suggested by Lombardi et al. (2005), the carboxylic groups of uronic acids are primarily responsible for binding copper by EPS derived from *Scenedesmus acuminatus* [8]. In the present study, a small amount of amino sugars was detected (0.42 and 0.38% in *P. kessleri* and *C. vulgaris* EPS, respectively). Although amino sugars are present mostly in cyanobacterial EPS [19], glucosamine has been detected also in EPS from *Dictyosphaerium chlorelloides* and *D. tetrachlorum* (0.8–1.6% wt.) [18].

The monosaccharide composition determined using TLC analysis showed the presence of rhamnose, xylose, mannose and galactose in EPS synthesized by *P. kessleri*, while rhamnose, mannose and galactose were detected in the EPS of *C. vulgaris* (Table 2).

The analysis of TLC plates showed that rhamnose was the dominant sugar in the studied EPS synthesized by *P. kessleri* and *C. vulgaris*. The results correspond to those reported by Ogawa (1999) and Capek et al. (2020) [6,16]. As demonstrated by El-Naggar et al. (2020), *C. vulgaris* polysaccharides are composed of glucose and rhamnose as the dominant EPS components with lower contents of fructose, maltose, lactose, and arabinose [17]. The differences in the sugar composition of exopolysaccharides, even those derived from the same species, indicate that a number of factors, e.g., culture conditions or the type of nutrition, can influence the structure of EPS. The composition of the EPS monosaccharide is important for elucidation of the EPS functions and application potential [15].

The ICP-OES analysis was performed to determine the content of essential elements in the EPS of the two analyzed microalgal species (control samples); its results are presented in Table 3. The obtained data indicate that the content of the elements in the EPS differed between the algal species. The Ca content was nearly 2-fold higher in the *C. vulgaris* EPS than in the *P. kessleri* EPS. Additionally, the EPS from *C. vulgaris* had higher contents of P and S. Higher Mg levels were observed in the EPS synthesized by *P. kessleri*. Other elements, including Na and K, did not incorporate into the EPS structure at amounts higher than 4 µg mL^−1^.

The studied EPS are characterized by high contents of Ca, Mg, S, and lower amounts of Fe, K, Na, Mn, and Zn in the EPS from both species. Jiao et al. (2010), who studied the metal composition in EPS extracted from biofilm growing on acidic mine drainage solutions, observed that the EPS metal composition was closely associated with the growth medium composition. The acidic mine drainage solutions contained mainly Fe and Al, and so did the extracted EPS [20]. In the BG-11 medium used in the current study, Ca and Mg are present in the highest concentration, which corresponds with the content of these metals in EPS. The presence of divalent ions, such as Ca (II) and Mg (II), was also detected in, for example, EPS produced by diatoms. These cations are involved in cross-linkages between the sugar molecules of EPS [21]. The analyzed EPS contained high amounts of sulfur. This element may incorporate into EPS in the form of sulfate groups [22]. It was reported that *C. vulgaris* synthesizes sulfated EPS. The content of SO_4_^2−^ residues in *C. vulgaris* EPS was determined by El-Naggar et al. (2020), who reported a sulfate content of 210.654 mg/g [17].

### 2.3. Metal Sorption

#### 2.3.1. ICP-OES Analysis

The ICP-OES analysis showed metal concentration-dependent differences in the cadmium and lead-binding capacity between the EPS from *P**. kessleri* and *C. vulgaris*. Greater differences in this parameter of EPS synthesized by *P**. kessleri* and *C. vulgaris* were found in the case of lead ion removal (Figure 1). The Pb ion removal potential of the *P. kessleri* EPS decreased with the increase in the metal concentration from 47% at 10 mg/L to 25.5% at 150 mg/L Pb (II). In the case of the EPS synthesized by *C. vulgaris* cells, the highest value of lead ion removal (49.3%) was achieved at 100 mg/L Pb, whereas the lowest removal was observed in the presence of 50 mg/L Pb (II). In the case of Cd (II) ions, the statistical analysis showed that both studied EPS removed cadmium at the same level at the metal concentration of 10–50 mg/L, and significant differences were observed only in the presence of Cd (II) at the concentration of 150 mg/L.

The calculation of the amount of adsorbed metal ions on the EPS in both species showed several times higher affinity of lead than cadmium for the studied EPS (Table 4). The highest cadmium accumulation levels were 35.35 and 48.67 mg/g, and the maximum lead sorption values were 264.1 mg/g and 573.6 mg/g for the EPS from *P. kessleri* and *C. vulgaris,* respectively. The EPS of *P. kessleri* showed a sorption maximum in the presence of both Cd (II) and Pb (II) at the concentration of 100 mg/L. The sorption capacity of EPS from *C. vulgaris* increased with the increasing metal ion concentration in the presence of both cadmium and lead ions. This may be related to the fact that this polymer has a greater number of binding sites that have not been saturated. EPS from *C. vulgaris* contain higher amounts of uronic acids and amino acids, which have carboxyl and amine groups, involved in the metal–EPS interaction [8].

The sorption capacity of the studied EPS toward Pb (II) was approximately 10-fold higher than in the case of Cd (II). The sorption results are in agreement with those obtained for EPS derived from bacteria *Paenibacillus jamilae*. The authors observed that EPS adsorbed 189.53 mg/L Pb (II) and 21.93 mg/L Cd (II) in the one-metal system containing 0.1 mM of metal ions [23]. Similar results were obtained for EPS synthesized by *Rhodococcus rhodococcus,* which adsorbed approx. 200 mg/g Pb (II) and approx. 50 mg/g Cd (II) [24]. The EPS sorption capacity depends not only on different environmental parameters, but also on the properties of heavy metals. The preference of EPS for lead ions may be correlated with differences in the hydration energy and the hydrated ionic radius of lead and cadmium ions. Due to their lower hydration energy than cadmium ions ((Pb (II) < Cd (II)), lead ions are more easily adsorbed, which contributes to quicker and more efficient adsorption) [25]. Reddad et al. (2003) suggested that cadmium ions have lower affinity to polysaccharide than lead [26].

The ICP-OES data indicated species-specific differences in the concentrations of elements released from EPS in the presence of cadmium and lead ions in comparison with the control samples (Table 5). Ca, Mg, and S were the main elements present in the control sample. A high concentration of these elements was also detected in the EPS composition. Therefore, the presence of the studied elements in the solution was the result of release from EPS. The presence of Cd and Pb ions resulted in a decrease in the amount of elements released by EPS from *P. kessleri*, in comparison to the control. An approximately Two-fold reduction in the Ca ion content in the solution was observed after cadmium sorption. In turn, in the presence of lead at concentrations of 10 mg/L and 50 mg/L, the release increased, and the reduction was observed only at the Pb (II) concentration of 150 mg/L. Increased Zn release was also noticed after the lead treatment at a concentration of 100 mg/L of *P. kessleri* EPS, compared to the control. In turn, the elemental analysis of the solution after metal sorption by *C. vulgaris* EPS showed a decrease in the release of Ca, Mg, and S ions (except Pb (II) at 50 mg/L), while the amount of P and Zn in the solution increased in the presence of both cadmium and lead ions.

It was observed that, in a multi-metal system, only Pb interacted with EPS at a similar level as in a one-metal system, whereas Cd was susceptible to the effects of the presence of other metal ions [23]. In the current study, EPS contained high amounts of Ca and Mg, which were released to the solution (Table 5); this may affect cadmium affinity to the exopolysaccharide.

#### 2.3.2. FTIR Analysis

The FTIR spectra of EPS produced by *P. kessleri* and *C. vulgaris* are presented in Figure 2. The EPS derived from *P. kessleri* and *C. vulgaris* produced similar FTIR spec-tra, although the *P. kessleri* EPS had a more heterogeneous 950–700 cm^−1^ region. Both EPS had bands at 2938 cm^−1^, 1730 cm^−1^ and 1416 cm^−1^. The EPS isolated from *P. kessleri* showed the presence of bands at 3317 cm^−1^, 1608 cm^−1^, 1507 cm^−1^, 1249 cm^−1^, 1046 cm^−1^, 906 cm^−1^, 860 cm^−1^, 815 cm^−1^, 776 cm^−1^, and 707 cm^−1^. In turn, the spectrum of EPS derived from *C. vulgaris* had bands at 3306 cm^−1^, 1597 cm^−1^, 1247 cm^−1^, 1035 cm^−1^, 834 cm^−1^, and 791 cm^−1^ (Figure 2, Table 6).

The analysis of the FTIR results showed that the spectra of both EPS contain bands typical for polysaccharides. The bands at approximately 3310 cm^−1^ and 2938 cm^−1^ are characteristic for stretching vibrations of -OH groups and -CH groups, respectively [27]. The 1750–1100 cm^−1^ region is characteristic for functional groups, such as esters, and the region below 1100 cm^−1^ is typical of ring and α and β linkage vibrations. The spectra of both samples show bands typical for esterified carboxyl groups (1730 cm^−1^) and asymmetric and symmetric vibrations of carboxylic ions COO^−^ (1608 cm^−1^ for *P. kessleri* and 1597 cm^−1^ for *C. vulgaris*, and 1416 cm^−1^, respectively). These carboxyl groups may originate from uronic acids [28]. The bands at 1249 and 1247 cm^−1^ in *P. kessleri* and *C. vulgaris,* respectively, may indicate the presence of phosphate groups [29] or sulfate groups [30]. The FTIR spectra of both EPS show the presence of stretching C-O and bending C-OH vibrations (1035 cm^−1^ and 1046 cm^−1^) [27,31]. The bands with lower wavenumbers (950–700 cm^−1^) are characteristic for pyranose and furanose rings as well as α- and β-glycosidic bonds [31,32]. The bands at 815 cm^−1^ and 906 cm^−1^ observed in the *P. kessleri* EPS may indicate the presence of mannose or mannuronic acid [30,33]. Mannan is also observed at 1147 cm^−1^ [30]. The band at 776 cm^−1^ was characterized as typical for guluronic acid [33] as well as the 791 cm^−1^ band in *C. vulgaris* EPS [34]. The band at 834 cm^−1^ found in the *C. vulgaris* EPS was assigned to a sulfated galactose unit [30]. The monosaccharides identified based on the FTIR spectra correspond to the results obtained from the TLC analysis, which also indicated the presence of mannose and galactose. In turn, the analysis of the elemental composition of EPS showed a higher concentration of sulfur than phosphorus; hence, the bands at 1247 and 1249 cm^−1^ may rather indicate sulfate groups.

**Table 6 molecules-27-07153-t006:** FTIR band assignment for the EPS of *P. kessleri* and *C. vulgaris*.

Species		Assignment	Interpretation	References
*P. kessleri*	*C. vulgaris*			
Wavenumber (cm^−1^)				
776, 815, 860	791, 834	(CO), δ(CH)	furanose and pyranose rings of saccharides α- and β-glycosidic bonds	[32] [31]
1046	1035	v(CO), δ(COH)	β(1,4) glycosidic bond	[31]
1249	1247	ν_as_(PO), ν_as_(SO)	phosphate group, sulfate group	[29] [30]
1416	1416	ν_s_(COO)	carboxymethyl groups	[31]
1608	1597	ν_as_(COO)	carboxymethyl groups	[31]
1730	1730	ν(C=O)	esterified carboxyl groups, uronic acid	[28] [31]
2919	2938	ν_as_(CH)	hydrocarbon bond	[27]
3358	3306	ν(OH)	hydroxyl groups	[27]

In the presence of Cd (II) ions, the band of O-H groups in the *C. vulgaris* and *P. kessleri* EPS shifted from approximately 3310 cm^−1^ to 3355 cm^−1^ and 3358 cm^−1^, respectively (Figure 2). The band at 2938 cm^−1^ shifted to a lower value of approximately 2920 cm^−1^. There was a band at 1623 cm^−1^, which was not observed in the control sample. Under the influence of Cd (II), the bands at around 1600 cm^−1^ and 1416 cm^−1^ were shifted to lower wavenumbers in the *P. kessleri* EPS but did not change in the case of the *C. vulgaris* EPS. The new band that appeared in the region 1400–1300cm^−1^ for the *C. vulgaris* EPS was shifted from 1372 cm^−1^ (Cd 10 mg/L) toward 1301 cm^−1^ (Cd 50 mg/L, Cd 100 mg/L, and Cd 150 mg/L). Moreover, bands at 858 cm^−1^, 815 cm^−1^, and 742 cm^−1^ were observed in the anomeric region.

The analysis of the FTIR spectra after Pb (II) sorption showed a shift of OH groups from 3306 cm^−1^ to 3354 cm^−1^ in the *C. vulgaris* EPS, while this band in the *P. kessleri* EPS did not change. The band of CH groups shifted from 2938 cm^−1^ to 2920 cm^−1^ and 2929 cm^−1^ in the EPS from *C. vulgaris* and *P. kessleri*, respectively. A band of esterified carboxylic groups (1730 cm^−1^) was observed only in the *C. vulgaris* EPS. In turn, a 1621 cm^−1^ band appeared in the *P. kessleri* EPS. In both samples, the bands of COO^−^ groups shifted from 1600 cm^−1^ and 1416 cm^−1^ to lower values, as in the presence of Cd (II) (Figure 2). In the anomeric region, bands at around 850 cm^−1^, and 788 cm^−1^ were observed. Additionally, a sharp band at 680 cm^−1^ was observed in *P. kessleri* EPS treated with Pb ions at a concentration of 50 mg/L and 100 mg/L.

The FTIR analysis of EPS after metal sorption showed the interaction of OH groups with Cd (II) ions in both samples and with Pb (II) in the case of the *P. kessleri* EPS. The shift in the OH band was also observed after Cd (II) treatment of *Bacillus cereus* EPS [35]. After the metal treatment in this study, the band at 1730 cm^−1^ disappeared, except for the *P. kessleri* EPS in the presence of Pb (II). This indicates the involvement of carboxylic esters in the binding of metal ions [28]. Moreover, the bands around 1600 cm^−1^ and 1416 cm^−1^ characteristic for COO^−^ observed in all tested samples shifted to lower wavenumbers, which indicates that carboxylic ions can also take part in ion binding [36]. Due to the use of cadmium and lead nitrates in the experiment, the bands in the region of 1372–1301 cm^−1^ may indicate antisymmetric stretching vibrations of N-O [37]. The anomeric region (900–750 cm^−1^) of the studied EPS was also affected by the addition of the heavy metal ions, and the bands identified for mannose, mannuronic acid, and guluronic acid were shifted or their intensity increased, which also evidenced that the ions were attached to the sugar unit. The band at 680 cm^−1^ observed in the EPS from *P. kessleri* in the presence of Pb (II) may indicate formation of lead oxide [38]. The metal–oxide interaction may be another explanation [39].

The FTIR spectroscopy results obtained in this study confirmed the presence of major functional carboxyl, carbonyl, and hydroxyl groups in the EPS of both analyzed species. The FTIR spectra of EPS produced by *C. vulgaris* and *P. kessleri* before and after the sorption indicate involvement of mainly the -COO^−^, -COH, and -OH groups in this process. Mota et al. (2013) reported that the carboxyl and hydroxyl groups played a more important role in the binding of metal to polysaccharides released by *Cyanothece* sp. CCY 0110 than other functional groups [40]. Based on results obtained after 2D-FTIR-COS analysis, Xie et al. (2020) reported that Cd (II) ions affect EPS in the following order: CO (carboxylic acid), COO^−^ (amino acids), C-OH (proteins), C-O-C, and C-OH (polysaccharides) [10].

The data obtained in the present study indicate that the higher sorption capacity of EPS synthesized by *C. vulgaris* in comparison to EPS from *P. kessleri* is related to the differences in the biochemical composition of both materials. EPS from *C. vulgaris* contain higher amounts of protein, amino acids, and uronic acids. The presence of e.g., uronic acids in EPS is responsible for their overall negative charge [8]. EPS containing negatively charged components such as uronic acids are characterized by a high metal-complexing capacity and are considered promising for the removal of toxic metals from contaminated environments [10].

The TLC results confirmed the presence of mannose in the EPS from both species analyzed in the present study. The shift of bands characteristic for the skeletal vibration region of pyranose and furanose of the main skeleton and sugar residues after the addition of heavy metal ions was also confirmed by the FTIR analyses performed in the current study. Allard and Casadevall (1990) reported that mannose is one of the sugars responsible for the formation of complexes with heavy metals, e.g., with lead [41].

## 3. Materials and Methods

### 3.1. Culture Strains and Pre-Cultivation

*Parachlorella kessleri* 250 and *Chlorella vulgaris* 898 was obtained from the Culture Collection of Autotrophic Organisms (Třeboň, Czech Republic). The inoculated algal cultures were grown in shaken Erlenmeyer flasks in sterilized liquid BG-11 medium at 60 µmol/m^2^/s light intensity and continuous light (24 h light/0 h dark) and aerated with sterile air.

### 3.2. Experimental Set-Up

The strains were grown in triplicate in 5000 mL glass bottles with a working volume of 2500 mL of sterilized BG 11. The batch cultures were maintained at the continuous light intensity of 60 µmol/m^2^/s at 21  ±  1 °C in a temperature-controlled room. The cultures were continuously aerated with sterile air and shaken at 90 rpm.

### 3.3. Isolation of EPS

After 22 days of cultivation, the cells and culture medium were separated by centrifugation at 9000 rpm for 30 min at 4 °C. Next, the cell-free supernatant was filtered under reduced pressure and concentrated using a rotary evaporator (20 mBar, 38 °C, Heidolph, Germany). An alcohol precipitation method was used for EPS isolation. EPS was precipitated from the supernatant by adding two volumes of ethanol (2:1 *v*/*v*), stirred, and left for 72 h at 4 °C. Next, the solutions were centrifuged at 9000 rpm for 30 min, dissolved in demineralized water, and dialyzed for 72 h at 4 °C. The EPS obtained was freeze-dried, redissolved in demineralized water, and left for 24 h at 4 °C. To obtain water-soluble EPS, the samples were dissolved in demineralized water and centrifuged, and the supernatant was lyophilized. Next, the samples were dissolved in demineralized water to a final concentration of 1 mg/mL and stored at 4 °C until analysis [42].

### 3.4. EPS Production

To determine the efficiency of EPS synthesis, the EPS yield and specific productivity were calculated. The total dry weight of biomass was calculated by the following formula:DW_t_ = DW · V(1)
where DW_t_ is the total dry weight of the cultured biomass [g]; DW is the biomass concentration [g/L]; and V is the culture volume [L].

Then, specific EPS productivity was calculated by
Specific EPS productivity [mg/g] = W_EPS_/DW_t_(2)
where W_EPS_ is the weight of lyophilized EPS [mg], and DWt is the total dry weight of cultured biomass [g].

The EPS yield was calculated per culture volume:EPS yield [mg/L] = W_EPS_/V(3)
where W_EPS_ is the weight of lyophilized EPS [mg], and V is the culture volume [L].

### 3.5. Analysis of Chemical Composition of EPS

#### 3.5.1. Biochemical Composition

The total sugar content was determined with the phenol–sulfuric method using glucose (POCH, Poland) as a standard [43]. The protein concentration was estimated with the Bradford method using bovine serum albumin (Sigma Aldrich, St. Louis, MO, USA) as a standard [44].

To determine the content of uronic acids, amino sugars, reducing sugars, and amino acids, samples of EPS were hydrolyzed. Water solutions of EPS were hydrolyzed using 4 M trifluoroacetic acid (TFA). The process was carried out at 100 °C for 4 h in a thermoblock (Macherey-Nagel, Duren, Germany) [42]. The samples were evaporated in a vacuum centrifuge, dissolved in demineralized water, and evaporated three times to remove TFA. After that, the hydrolyzed samples were dissolved to a final concentration of 1 mg/mL and stored at 4 °C until analysis. The content of reducing sugars, uronic acids, and amino sugars was determined using the Somogyi–Nelson [45], carbazole–sulfuric [46], and modified Elson–Morgan methods [47], respectively, with glucose (Sigma Aldrich, St. Louis, MO, USA), galacturonic acid (Sigma Aldrich, St. Louis, MO, USA), and D-glucosamine (Glentham Life Science, Corsham, UK) as standards, respectively. The concentration of amino acids was measured with the ninhydrin method [48] using glycine (POCH, Gliwice, Poland) as a standard.

#### 3.5.2. Monosaccharide Composition

The sugar composition was analyzed by thin layer chromatography (TLC) using silica gel-coated TLC plates (Merck, Darmstadt, Germany). Glucose, galactose, mannose, fructose, xylose and rhamnose (Sigma Aldrich, St. Louis, MO, USA) at a concentration of 1 mg/mL were used as standards. Then, 30 µL of hydrolyzed EPS and standard solutions were transferred on TLC plates (Merck, Darmstadt, Germany) and dried. The plates were placed in a chromatographic chamber, which was first saturated by the developing phase containing 1-propanol:ethyl acetate:water 4:0.5:0.5 *v/v* for 2 h. Chromatograms were developed for 4 h; then, the plates were dried and sprinkled with 10% H_2_SO_4_ in ethanol and heated at 100 °C for 15 min to visualize separated compounds [49].

#### 3.5.3. Elemental Composition

Additionally, the elemental composition of EPS was examined to evaluate the sorption capacity toward ions contained in the growth medium. For this purpose, lyophilized samples were dissolved in 5% HNO_3_ to a final EPS concentration of 100 mg/L. The measurements were performed using the ICP-OES method.

### 3.6. Metal Sorption

#### 3.6.1. Preparation of Solutions and Sorption Experiment

For the experiment, the following concentrations of Pb(NO_3_)_2_ and Cd(NO_3_)_2_∙4H_2_O were used: 10, 50, 100, and 150 mg/L. Samples for the sorption process were prepared by dissolving EPS in lead and cadmium nitrate solutions to a final EPS concentration of 100 mg/L. The procedure preparing sorption experiment was according to Dobrowolski et al. (2017) with some modifications [24]. The pH value of the solutions was adjusted to 5. The samples were agitated at 120 rpm for 30 min at 25 °C. Then, the supernatant was centrifuged at 9500 rpm for 12 min (Rotanta 460 RS, Hettich, Kirchlengern, Germany). The supernatant was filtered using a hydrophilic 0.22 µm PES filter (Chemland, Krakow, Poland). The control samples were prepared according to the same procedure, but the EPS were dissolved in water instead of metal solutions. After the sorption experiment, the filtrates were used for measurement of metal ion concentrations with inductively coupled plasma iCAP 6500 Duo (ICP-OES, Thermo Fisher Scientific, Waltham, MA, USA). The pellet was freeze-dried for FTIR analysis. The sorption experiments were performed in triplicate.

#### 3.6.2. ICP-OES

To evaluate the ability of EPS to bind metals, the optical emission spectrometry with inductively coupled plasma analysis was employed in this study. The samples were measured with the use of ICP-OES iCAP 6500 Duo (Thermo Fisher Scientific, Waltham, MA, USA) equipped with a charge injection device (CID) detector and TEVA software. CCS-6 obtained from Inorganic Ventures (Christiansburg, VA, USA) was used as a standard solution for the determination of elements (100 µg/mL in 7% HNO_3_, Inorganic Ventures, Christiansburg, VA, USA). The device worked at the following parameters: RF generator power 1150 W, RF generator frequency 27.12 MHz, carrier gas flow rate 0.65 dm^3^/min, coolant gas flow rate 16 dm^3^/min, and auxiliary gas 0.4 dm^3^/min. The wavelength was 220.353 nm for the determination of the lead content and 214.438 nm for cadmium [50]. The following formula was used for the calculation of the percentage removal potential:Removal potential [%] = (c_i_ − c)/ci ∙ 100 % (4)

The sorption capacity was calculated as follows:Sorption capacity [mg/g] = ((c_i_ − c) ∙ V)/W(5)
where c_i_ is the initial concentration of metal ions [mg/L]; c is the concentration of metal ions after incubation with EPS [mg/L]; V is the sample volume [L]; and W is the weight of EPS in the sample [g] [51].

The content of Ca, Mg, Mn, P, S and Zn in the solution after the sorption process was determined as well.

#### 3.6.3. Fourier Transform Infrared Spectroscopy

Fourier transform infrared spectroscopy (FTIR) spectra were collected via a Nicolet 6700 FTIR spectrometer (Thermo Scientific, Waltham, MA, USA). The Smart iTR attenuated total reflection (ATR) sampling accessory was used. Freeze-dried samples were placed directly on the ATR crystal and measured. The spectra were collected over the range of 4000–650 cm^−1^. For each sample, 200 scans at a spectral resolution of 4 cm^−1^ were averaged. These spectra were normalized to 1.0 at 1030–1046 cm^−1^. All spectral manipulations were carried out using Origin Pro 8.5 (OriginLab Corporation, Northampton, MA, USA).

### 3.7. Statistical Analysis

All experiments were performed in triplicate, and the mean value of the data was reported.

The significance of the differences in the measured parameters was determined using one-way ANOVA and Tukey’s post-hoc test at *p* < 0.05, STATISTICA 12 (Statsoft, Inc., Tulsa, OK, USA).

## 4. Conclusions

In order to determine the potential of the extracellular polymeric substances produced by *P. kessleri* and *C. vulgaris* to be used as biosorbents for heavy metal removal, their biochemical characteristics and the Pb (II) and Cd (II) sorption capacity were investigated. The productivity of EPS achieved 16.18 and 12.73 mg/L, respectively for *P. kessleri* and *C. vulgaris*. The analysis of the biochemical composition of the obtained EPS showed that the main component of both polymers are carbohydrates. EPS synthesized by *C. vulgaris*, although containing fewer carbohydrates, has a higher content of reducing sugars and uronic acids, as well as proteins, amino acids and amino sugars by 23.4%, 8.3%, 26.7%, 59.3% and 4.5%, respectively. Rhamnose, mannose and galactose were identified in the studied EPS, and xylose was also found in the EPS produced by *P. kessleri*. The presence of mannose and galactose was confirmed also by the FTIR spectrum.

The ICP-OES analysis of the elemental composition showed high contents of calcium, magnesium, and sulfur in the EPS from *P. kessleri* and *C. vulgaris*. The polymer produced by *C. vulgaris* showed higher sorption capacity than EPS from *P. kessleri*. For cadmium, the sorption capacity was 45.9% higher, and for lead, 54.1%, in the presence of metal ions of 150 mg/L. The results of the FTIR analysis indicate the involvement of the carboxyl, hydroxyl, and carbonyl groups as binding sites for divalent cations of the analyzed heavy metals on the surface of EPS. The results obtained in this study indicate that, due to the significant amount of negatively charged uronic acids and amino acids, the EPS produced by *C. vulgaris* have potential as biosorbents in water and wastewater bioremediation.

## Figures and Tables

**Figure 1 molecules-27-07153-f001:**
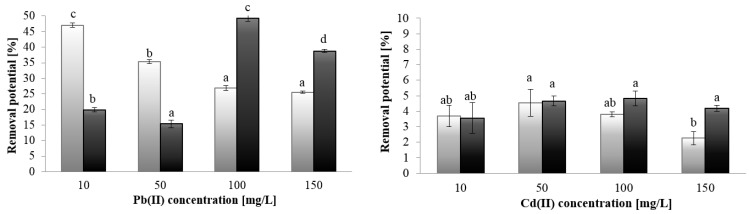
Removal potential of EPS from *P. kessleri* (light grey bars) and *C. vulgaris* (dark grey bars) towards Pb (II) and Cd (II). The letters (a, ab, b, c, d) on the graphs indicate statistical significance of the presented results analyzed by Tukey test (*p* ≥ 0.05).

**Figure 2 molecules-27-07153-f002:**
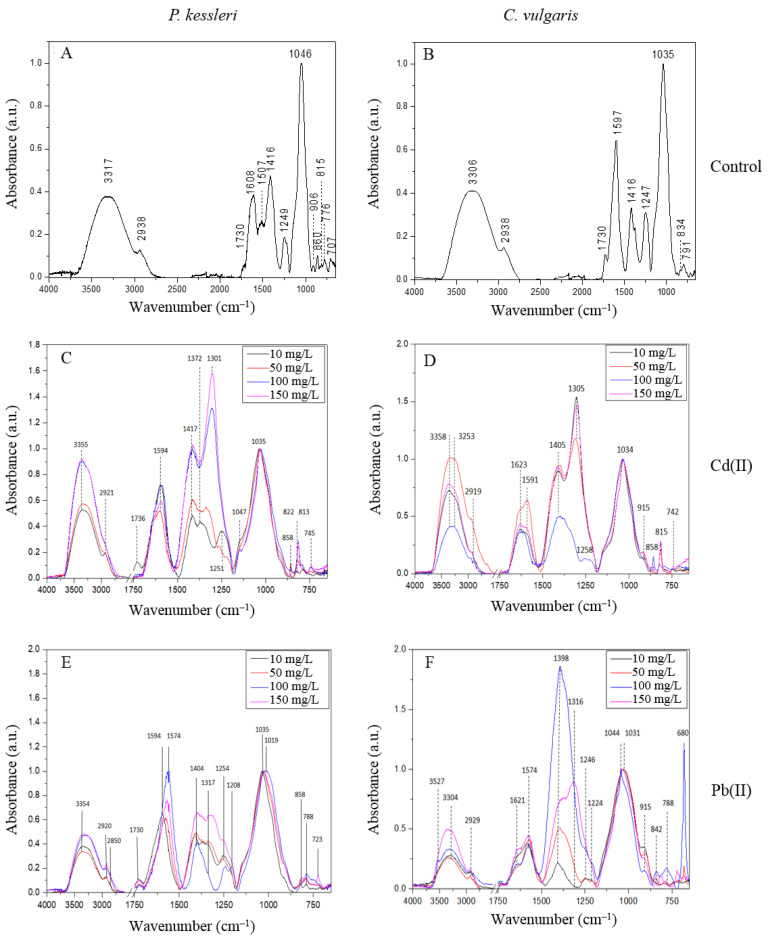
FTIR spectra of EPS from *P. kessleri* (**A**,**C**,**E**) and EPS from *C. vulgaris* (**B**,**D**,**F**) before (**A**,**B**) after additions of cadmium (**C**,**D**) and lead (**E**,**F**) in the range of 3750-650 cm^−1^ (the region 2750–1750 cm^−1^ was removed due to the lack of spectral features).

**Table 1 molecules-27-07153-t001:** The biomass yield and the productivity and biochemical composition of EPS synthesized by *P. kessleri* and *C. vulgaris.* The superscript letters (a, b) in the table indicate statistical significance of the presented results analyzed by ANOVA Tukey’s test (*p* ≥ 0.05) (EPS = 1 mg/mL; *n* = 3; ±SD).

	*P. kessleri*	*C. vulgaris*
Biomass yield [g/L]	0.77 ± 0.08 ^a^	0.86 ± 0.09 ^a^
EPS yield [mg/L]	12.49 ± 1.21 ^b^	10.42 ± 1.20 ^a^
EPS specific productivity [mg/g DW]	16.18 ± 1.09 ^b^	12.73 ± 2.09 ^a^
Compound [µg/mg EPS]		
Carbohydrates ^1^	635.0 ± 15.3 ^a^	577.3 ± 19.34 ^b^
Proteins ^1^	5.5 ± 0.41 ^b^	7.5 ± 0.15 ^a^
Reducing sugars ^2^	158.2 ± 7.30 ^a^	206.4 ± 4.17 ^b^
Uronic acids ^2^	123.0 ± 2.46 ^a^	134.1 ± 5.02 ^b^
Amino acids ^2^	14.4 ± 0.84 ^a^	35.4 ± 1.32 ^b^
Amino sugars ^2^	4.2 ± 0.20 ^a^	4.4 ± 0.32 ^a^

^1^ Non-hydrolyzed EPS; ^2^ Hydrolyzed EPS.

**Table 2 molecules-27-07153-t002:** Monosaccharide composition of EPS synthesized by *P. kessleri* and *C. vulgaris*. Retention factor (Rf) is the distance travelled by an individual component divided by the total distance travelled by the solvent.

R_f_	*P. kessleri*	*C. vulgaris*
Rha	0.77	0.80
Xyl	0.68	-
Man	0.56	0.56
Gal	0.43	0.41

**Table 3 molecules-27-07153-t003:** Elemental composition of *P. kessleri* and *C. vulgaris* EPS (µg/mg) (±SD; *n* = 3).

Element	*P. kessleri*	*C. vulgaris*
Ca	45.30 ± 0.55	86.73 ± 0.09
Mg	34.12 ± 0.17	20.02 ± 0.10
Mn	3.49 ± 0.02	0.76 ± 0.00
P	4.85 ± 0.58	18.00 ± 0.91
S	27.16 ± 0.11	44.61 ± 0.14
Zn	2.34 ± 0.01	0.54 ± 0.01

**Table 4 molecules-27-07153-t004:** Sorption capacity of EPS synthesized by *P. kessleri* and *C. vulgaris* toward Cd (II) and Pb (II). The superscript letters (a, ab, b, c, and d) in the table indicate statistical significance of the presented results (C_EPS_ = 100 mg/g; ANOVA, Tukey’s test, *p* ≥ 0.05) (±SD).

Metal Ion Concentration [mg/L]	Sorption Capacity [mg/g]			
	Cd (II)		Pb (II)	
	*P. kessleri*	*C. vulgaris*	*P. kessleri*	*C. vulgaris*
10	3.85 ± 1.06 ^c^	3.73 ± 1.07 ^a^	50.96 ± 1.55 ^a^	20.74 ± 0.85 ^a^
50	15.37 ± 2.34 ^a^	22.73 ± 1.33 ^b^	157.27 ± 1.59 ^b^	79.43 ± 2.38 ^b^
100	35.35 ± 3.18 ^b^	41.1 ± 4.15 ^c^	264.1 ± 10.32 ^c^	490.85 ± 1.06 ^c^
150	26.33 ± 4.93 ^ab^	48.67 ± 2.08 ^d^	263.0 ± 20.0 ^c^	573.6 ± 10.61 ^d^

**Table 5 molecules-27-07153-t005:** The concentration of elements released from EPS to solution after cadmium and lead sorption in comparison to control sample (EPS dissolved in demineralized water) (±SD).

Element [mg/mL]	Cd (II) [mg/L]				Pb (II) [mg/L]				Control
	10	50	100	150	10	50	100	150	
	EPS *P. kessleri*								
Ca	3.06 ± 0.02	2.65 ± 0.03	3.74 ± 0.02	2.85 ± 0.02	7.98 ± 0.04	7.47 ± 0.02	7.05 ± 0.05	5.36 ± 0.04	7.12 ± 0.13
Mg	2.53 ± 0.01	2.54 ± 0.02	3.56 ± 0.02	2.92 ± 0.01	3.50 ± 0.01	3.13 ± 0.01	3.18 ± 0.01	3.39 ± 0.02	3.83 ± 0.04
Mn	0.02 ± 0.00	0.01 ± 0.00	0.02 ± 0.00	0.01 ± 0.00	0.05 ± 0.00	0.05 ± 0.00	0.04 ± 0.00	<0.01	0.55 ± 0.01
P	0.91 ± 0.03	0.70 ± 0.01	0.73 ± 0.04	0.57 ± 0.07	0.29 ± 0.04	0.36 ± 0.08	0.18 ± 0.10	0.02 ± 0.01	1.94 ± 0.08
S	1.65 ± 0.01	1.39 ± 0.01	2.16 ± 0.02	1.81 ± 0.02	2.45 ± 0.01	1.96 ± 0.01	1.31 ± 0.01	2.20 ± 0.01	3.21 ± 0.06
Zn	0.58 ± 0.00	0.49 ± 0.01	0.34 ± 0.00	0.36 ± 0.00	0.22 ± 0.00	0.65 ± 0.00	0.93 ± 0.00	0.13 ± 0.00	0.69 ± 0.06
	EPS *C. vulgaris*								
Ca	6.58 ± 0.01	5.86 ± 0.05	6.47 ± 0.02	6.42 ± 0.01	4.05 ± 0.03	8.39 ± 0.04	5.49 ± 0.08	5.63 ± 0.07	7.94 ± 0.02
Mg	1.53 ± 0.00	1.45 ± 0.01	1.63 ± 0.01	1.60 ± 0.00	1.01 ± 0.01	2.20 ± 0.01	1.29 ± 0.10	1.16 ± 0.02	1.96 ± 0.03
Mn	0.06 ± 0.00	0.05 ± 0.00	0.05 ± 0.00	0.05 ± 0.00	0.03 ± 0.00	0.06 ± 0.00	0.04 ± 0.00	0.04 ± 0.00	<0.01
P	1.05 ± 0.06	0.90 ± 0.05	0.93 ± 0.12	0.81 ± 0.09	0.60 ± 0.03	0.73 ± 0.16	0.61 ± 0.09	0.65 ± 0.04	0.01 ± 0.00
S	3.08 ± 0.01	2.69 ± 0.02	2.85 ± 0.02	2.96 ± 0.01	1.94 ± 0.01	3.92 ± 0.05	2.25 ± 0.05	1.50 ± 0.07	3.17 ± 0.02
Zn	0.70 ± 0.00	0.86 ± 0.00	0.56 ± 0.01	1.04 ± 0.01	0.84 ± 0.00	1.09 ± 0.00	1.51 ± 0.05	0.55 ± 0.01	0.20 ± 0.00

## Data Availability

Not applicable.

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
