# Peer review of "Soluble Extracellular Polymeric Substances Produced by Parachlorella kessleri and Chlorella vulgaris: Biochemical Characterization and Assessment of Their Cadmium and Lead Sorption Abilities"

_molecules, 2022, doi:10.3390/molecules27217153_

Round 1
Reviewer 1 Report
Comments to the manuscript
1. Abstract, line 5 from the top: "The composition of monosaccharides exhibited the presence of rhamnose, mannose and galactose, in the composition of EPS of both species." Comment: The sentence requires correction.
2. Abstract, line 6 from the top: "...The ICP-OES analyses..." Comment: When first used, the acronym should appear along with the full name.
3. Abstract, lines 7/8 from the top: "The sorption capacity of C. vulgaris EPS increased with the increase in the amount of metal ions." and page 5, the sentence above table 4: "The sorption capacity of EPS from C. vulgaris increased with the increasing metal ion concentration in the presence of both cadmium and lead ions." Comment: This relationship requires some explanation.
4. Page 1, line 2 from the bottom Comment: "Pb" as symbol for lead is a specific acronym and is usually written without parentheses.
5. Page 2, lines 13/15 from the top: "...Limited information is available on the exopolysaccharides synthesized by unicellular Chlorophyta, their bio-chemical characteristics, and sorption abilities [8–12]..." Comment: The authors mention a limited amount of information, and at the same time provide 5 references in this sentence?
6. Page 2, lines 25/28 from the top: "... Another objective was to evaluate the cadmium and lead sorption abilities of EPS based on the analysis of changes in functional groups in EPS induced by these metals performed with the use of Fourier transform infrared spectroscopy (FTIR) and optical emission spectrometry with inductively coupled plasma (ICP-OES)..." Comment: The sentence needs to be corrected.
7. Page 5, lines 2/3 from the top: "...The letters on the graphs indicate differences in the statistical significance of the presented results (Tukey test, p ≥0.05)...." Comment: The sentence is not clear.
8. Page 5, lines 6/7: "...The highest cadmium accumulation levels were 35.35 and 48.67 μg mg-1, and the maximum lead sorption values were 264.1 mg g-1 and..." Comment: Uniformity of units needed.
9. Do Figure 1 and Table 4 contain the same data?
10. Page 10, lines 8/10 from the top: "...In this study, only water-soluble EPS, which was obtained in another centrifugation and freeze-drying step, was analyzed [42]...." Comment: The sentence is not clear.
11. Page 10, line 16 from the bottom: "...The process was carried out at 100°C for 4 hours in a thermoblock...." Comment: Information on a thermoblock supplier is required.
12. Page 10, lines 12/14 from the bottom: "...After that, the hydrolyzed samples were dissolved to a final concentration of 1 mg mL-1 and stored at 4°C until analysis [42]...." Comment: Reference [42] should be placed elsewhere.
13. Page 11, lines 18/19: "...The supernatant was filtered using a hydrophilic 0.22 μm PES filter (Chemland, Poland)...." Comment: Possibly the use of such a filter causes ion losses during filtering by binding them to the filter?
14. Errors in the list of references, for example, item 39.
15. References: 44, 46, 47, 48 and 49. Comment: Consider using more recent sources.
16. I suggest that the authors consider replacing the units: "mg L-1", "mg mL-1", "m2 s-1", "cm3 g-1", "mg g-1", dm3 min-1 with units: "mg/L", mg/mL", "m2/s", "cm3/g", "mg/g", dm3/min, respectively, as easier to read.

Author Response
First of all, I would like to thank the Reviewers for the reviews. I would like to say that I am immensely grateful to the Reviewers for all remarks.
All comments on the manuscript have been considered and improvements have been applied.
All changes are marked in the manuscript.
ANSWERS TO THE COMMENTS OF REVIEWER
Comment 1.
"The composition of monosaccharides exhibited the presence of rhamnose, mannose and galactose, in the composition of EPS of both species. The sentence requires correction.”
Answer: The sentence has been corrected:
“The analysis of the monosaccharide composition showed the presence of rhamnose, mannose, and galactose in the EPS obtained from both species.”
Comment 2.
"...The ICP-OES analyses...When first used, the acronym should appear along with the full name.”
Answer: The acronym has been expanded:
“The ICP-OES (Inductively Coupled Plasma Optical Emission Spectrometry) analyses demonstrated that C. vulgaris EPS showed higher sorption capacity in comparison to P. kessleri EPS.”
Comment 3.
"The sorption capacity of C. vulgaris EPS increased with the increase in the amount of metal ions." and page 5, the sentence above table 4: "The sorption capacity of EPS from C. vulgaris increased with the increasing metal ion concentration in the presence of both cadmium and lead ions. This relationship requires some explanation.”
Answer: The relationship has been explained:
“The sorption capacity of EPS from C. vulgaris increased with the increasing metal ion concentration in the presence of both cadmium and lead ions. This may be related to the fact that this polymer has a greater number of binding sites that have not been saturated. EPS from C. vulgaris contains higher amounts of uronic acids and amino acids with carboxyl and amine groups, which are mostly involved in the metal-EPS interaction [8].”
Comment 4.
"Pb" as symbol for lead is a specific acronym and is usually written without parentheses.”
Answer: The “Pb” symbol has been removed.
Comment 5.
"...Limited information is available on the exopolysaccharides synthesized by unicellular Chlorophyta, their bio-chemical characteristics, and sorption abilities [8–12]... The authors mention a limited amount of information, and at the same time provide 5 references in this sentence?”
Answer:
The sentence has been rephrased.
“The properties of EPS depend largely on their chemical composition. Lombardi et al. (2005) indicated that the carboxyl groups of EPS are mostly involved in copper complexation [8,9]. Additionally, proteins were found to be important in metal sorption by EPS produced by C. vulgaris (Cd and Ag sorption) and Chlorella pyrenoidisa (As sorption) [10–12]. However, these studies do not determine the content of uronic acids, amino acids, and amino sugars, which also have functional groups that can interact with metal ions.”
Comment 6.
"... Another objective was to evaluate the cadmium and lead sorption abilities of EPS based on the analysis of changes in functional groups in EPS induced by these metals performed with the use of Fourier transform infrared spectroscopy (FTIR) and optical emission spectrometry with inductively coupled plasma (ICP-OES)... The sentence needs to be corrected.”
Answer:
The sentence has been corrected:
“Additionally, the cadmium and lead sorption ability of the obtained EPS was studied. For this purpose, the amount of metal ions bound to the EPS was determined using ICP-OES. Fourier transform infrared spectroscopy (FTIR) was used to identify functional groups involved in the EPS-metal interactions.”
Comment 7.
"...The letters on the graphs indicate differences in the statistical significance of the presented results (Tukey test, p ≥0.05).... The sentence is not clear.”
Answer:
This sentence has been corrected:
“The superscript letters (a, ab, b, c, d) on the graphs indicate statistical significance of the presented results analysed using the Tukey test (p ≥0.05).”
Comment 8.
"...The highest cadmium accumulation levels were 35.35 and 48.67 μg mg-1, and the maximum lead
sorption values were 264.1 mg g-1 and... Uniformity of units needed.”
Answer: The units have been unified:
Comment 9.
“Do Figure 1 and Table 4 contain the same data?”
Answer:
Figure 1 presents the removal potential, which was calculated according to equation 4.
Table 4 shows the results of sorption capacity, which was calculated using equation 5.
Comment 10.
"...In this study, only water-soluble EPS, which was obtained in another centrifugation and freeze-drying step, was analyzed [42].... The sentence is not clear.”
Answer:
The sentence has been corrected:
“To obtain water-soluble EPS, the samples were dissolved in demineralized water and centrifuged and the supernatant was lyophilized. Next, the samples were dissolved in demineralized water to a final concentration of 1 mg/mL and stored at 4⁰C until analysis [42].”
Comment 11.
"...The process was carried out at 100°C for 4 hours in a thermoblock.... Information on a thermoblock supplier is required.”
Answer:
Required information on a thermoblock supplier has been added:
Comment 12.
"...After that, the hydrolyzed samples were dissolved to a final concentration of 1 mg mL-1 and
stored at 4°C until analysis [42].... Reference [42] should be placed elsewhere.”
Answer:
Reference [42] has been relocated.
Comment 13.
"...The supernatant was filtered using a hydrophilic 0.22 μm PES filter (Chemland, Poland).... Possibly the use of such a filter causes ion losses during filtering by binding them to the filter?”
Answer:
Thank you for your comment. Before deciding on the type of filter, measurements comparing PES and PTFE filters were performed. For this purpose, an experiment of lead sorption by EPS was carried out (as in the manuscript). The results obtained were within statistical error.
Comment 14.
“Errors in the list of references, for example, item 39.”
Answer:
The Reference list has been corrected:
- Halaj, M.; Chválová, B.; Cepák, V.; Lukavský, J.; Capek, P. Searching for Microalgal Species Producing Extracellular Biopolymers. Chemical Papers2018, 72, 2673–2678doi:10.1007/s11696-018-0517-4.
- Ogawa, K.; Ikeda, Y.; Kondo, S. A New Trisaccharide, a-D-Glucopyranuronosyl-(1→3)- a-L-Rhamnopyranosyl-(1→ 2)-a-L-Rhamnopyranose From Chlorella vulgaris. Carbohydrate Research1999,321, 128-131.
- Perez, J.A.M.; Garcıa-Ribera, R.; Quesada, T.; Aguilera, M.; Ramos-Cormenzana, A.; Monteoliva-Sanchez, M. Biosorption of Heavy Metals by the Exopolysaccharide Produced by Paenibacillus jamilae. World J Microbiol Biotechnol2008, 24, 2699-2704 doi: 10.1007/s11274-008-9800-9
- Reddad, Z.; Ge, C.; Cloirec, P.L. Cadmium and Lead Adsorption by a Natural Polysaccharide in MF Membrane Reactor: Experimental Analysis and Modelling. Water Research2003, 37, 3983-3991 doi:10.1016/S0043-1354(03)00295-1
- Trivedi, M.K.; Dahryn Trivedi, A.B. Spectroscopic Characterization of Disodium Hydrogen Orthophosphate and Sodium Nitrate after Biofield Treatment. J Chromatogr Sep Tech2015, 06, 5 doi:10.4172/2157-7064.1000282.
- Senvaitiene, J.; Smirnova, J.; Beganskiene, A.; Kareiva, A. XRD and FTIR Characterisation of Lead Oxide-Based Pigments and Glazes. Acta Chim. Slov.2007, 54, 185–193
- Bailey, J.L. Techniques in Protein Chemistry; Elsevier Publishing Company, 1967; ISBN 978-0-444-40369-8.
Comment 15.
“References: 44, 46, 47, 48 and 49. Consider using more recent sources.”
Answer:
Thank you for your suggestion. These publications are original references for the analytical methods used and described in this manuscript. Most of them are still cited, especially the Bradford and DuBois methods. Reference 49 is specific, as it describes the composition of the developing phase used in the TLC method. Examples of recent publications citing the references mentioned:
Czemierska, M.; Szcześ, A.; Hołysz, L.; Wiater, A.; Jarosz-Wilkołazka, A. Characterisation of Exopolymer R-202 Isolated from Rhodococcusrhodochrous and Its Flocculating Properties. European Polymer Journal, 2017, 88, 21–33, doi:10.1016/j.eurpolymj.2017.01.008. [43-48]
Hanaka, A., Nowak, A., Ozimek, E., Dressler, S., Plak, A., Sujak, A., Reszczyńska, E., Strzeski M., Effect of copper stress on Phaseoluscoccineus in the presence of exogenous methyl jasmonate and/or Serratiaplymuthica from the Spitsbergen soil, Journal of Hazardous Materials, 2022, 436, 129232https://doi.org/10.1016/j.jhazmat.2022.129232 [45]
Asemaninejad, A., Langley, S., Mackinnon T., Spiers, G., Beckett, P., Mykytczuk, N., Basiliko, N., Blended municipal compost and biosolids materials for mine reclamation: Long-term field studies to explore metal mobility, soil fertility and microbial communities, Science of the Total Environment, 2021, 760, 143393 https://doi.org/10.1016/j.scitotenv.2020.143393 [45]
Yan, J.-K., Chen, T.-T., Wang, Z.-W., Wang C., Liu, C., Li, L., Comparison of physicochemical characteristics and biological activities of polysaccharides from barley (Hordeum vulgare L.) grass at different growth stages, Food Chemistry, 2022, 389, 133083 https://doi.org/10.1016/j.foodchem.2022.133083 [43, 44, 46]
Son, Y-J., Hwang, I.-K., Nho, C.W., Kim, S.M., Kim., S.H., Determination of Carbohydrate Composition in Mealworm (Tenebrio molitor L.) Larvae and Characterization of Mealworm Chitin and Chitosan, Foods, 2021, 10, 640 [43, 47]
Comment 16
I suggest that the authors consider replacing the units: "mg L-1", "mg mL-1", "m2 s-1", "cm3 g-1","mg g-1", dm3 min-1 with units: "mg/L", mg/mL", "m2/s", "cm3/g", "mg/g", dm3/min, respectively, as easier to read.
Answer:
The units have been replaced.

Reviewer 2 Report
This manuscript deals with preparing polymeric substances produced by Parachlorella Kessleri and Chlorella Vulgarisa and their application for the adsorption of cadmium and lead. The authors found that the contents of reducing sugars, uronic acids, and amino acids were higher in EPS synthesized by C. vulgaris than in EPS from P. kessleri. The ICP-OES analyses demonstrated that C. vulgaris EPS showed higher sorption capacity than P. kessleri EPS. The sorption capacity of C. vulgaris EPS increased with the amount of metal ions. P. kessleri EPS had a maximum sorption capacity in the presence of 100 mg L-1 of metal ions. The authors found that the %R for lead ion was 49.3%. In my opinion, this work is incomplete, and significant issues require careful revision before publication. Specifically:
Section 2. Results:
1.1. Metal Sorption: I think this part needs more work; the authors should study different factors which could affect the adsorption efficiency like pH, contact time, and amount of adsorbent and after that, find the optimum conditions and they should do this to increase the %R of each metal ion.
1.2. For the analysis part, IR is not enough to confirm the presence of metal ions on the surface. The authors can do Raman, SEM, and EDX before and after adsorption to determine the amount of metal ion adsorbed on the surface.
Section 4. Conclusion: Please rewrite the conclusion and include the obtained results in it.
Author Response
First of all, I would like to thank the Reviewers for the reviews. I would like to say that I am immensely grateful to the Reviewers for all remarks.
All comments on the manuscript have been considered and improvements have been applied.
All changes are marked in the manuscript.
ANSWERS TO THE COMMENTS OF REVIEWER
Comment 1.
Metal Sorption: I think this part needs more work; the authors should study different factors which could affect the adsorption efficiency like pH, contact time, and amount of adsorbent and after that, find the optimum conditions and they should do this to increase the %R of each metal ion.
Answer:
Thank you for your comment. The amount of material currently available is insufficient to perform sorption experiments in terms of pH, contact time and, in particular, EPS concentration. The time required to prepare the samples and carry out is approximately 2 months. This is significantly longer than the time taken to submit responses to reviews. However, we will include suggestions in the next publication.
Comment 2.
For the analysis part, IR is not enough to confirm the presence of metal ions on the surface. The authors can do Raman, SEM, and EDX before and after adsorption to determine the amount of metal ion adsorbed on the surface.
Answer:
Thank you for your suggestions. The analyses proposed provide more detailed information on the metal-EPS interactions. However, the aim of the FTIR studies was to identify the functional groups and visualize the changes which occur in the presence of cadmium and lead ions. This is the technique used in the literature cited below. We will include the proposed analyses in our future projects.
Mathivanan, K.; Chandrika, J.U.; Mathimani, T.; Rajaram, R.; Annadurai, G.; Yin, H. Production and Functionality of Exopolysaccharides in Bacteria Exposed to a Toxic Metal Environment. Ecotoxicology and Environmental Safety 2021, 208, 111567 doi:10.1016/j.ecoenv.2020.111567
Zhang, P.; Chen, Y.-P.; Peng, M.-W.; Guo, J.-S.; Shen, Y.; Yan, P.; Zhou, Q.-H.; Jiang, J.; Fang, F. Extracellular Polymeric Substances Dependence of Surface Interactions of Bacillus subtilis with Cd2+ and Pb2+: An Investigation Combined with Surface Plasmon Resonance and Infrared Spectra. Colloids and Surfaces B: Biointerfaces 2017, 154, 357–364, doi:10.1016/j.colsurfb.2017.03.046.
Ye, S., Zhang, M., Yang, H., Wang, H., Xiao, S., Liu, Y., Wang, J., Biosorption of Cu2+, Pb2+and Cr6+by a novel exopolysaccharide fromArthrobacter ps-5, Carbohydrate Polymers, 2014, 101, 50-56
Comment 3
Please rewrite the conclusion and include the obtained results in it.
Answer:
The conclusions have been rewritten including obtained results.
“In order to determine the potential of the extracellular polymeric substances produced by P. kessleri and C. vulgaris to be used as biosorbents for heavy metal removal, their biochemical characteristics and the Pb(II) and Cd(II) sorption capacity were investigated. The productivity of EPS achieved 16.18 and 12.73 mg/L, respectively for P. kessleri and C. vulgaris. The analysis of biochemical composition of the obtained EPS showed, that the main component of both polymers are carbohydrates. EPS synthesized by C. vulgaris, although containing fewer carbohydrates, has a higher content of reducing sugars and uronic acids, as well as proteins, amino acids and amino sugars by 23.4%, 8.3%, 26.7%, 59.3% and 4.5% respectively. Rhamnose, mannose and galactose were identified in studied EPS, and xylose was also found in EPS produced by P. kessleri. The presence of mannose and galactose was confirmed also by the FTIR spectrum.
The ICP-OES analysis of the elemental composition showed high content of Ca, Mg, and S in the studies EPS. The polymer produced by C. vulgaris showed higher sorption capacity than EPS from P. kessleri. For cadmium, the sorption capacity was 45.9% higher, and for lead 54.1%, in the presence of metal ions of 150 mg/L. The results of the FTIR analysis indicated involvement of carboxyl, hydroxyl, and carbonyl groups as binding sites for divalent cations of the analyzed heavy metals on the surface of EPS. The results obtained in this study indicated that, due to the significant amount of negatively charged carboxyl groups of uronic acids and amino acids, the EPS produced by C. vulgaris have potential to be used as a biosorbent in water and wastewater bioremediation.”

Reviewer 3 Report
The paper contributes to the journal. Just explan why o pH 5 in the Preparation of Solutions and Sorption Experiment, and add the range of solutions of ICP.
Author Response
First of all, I would like to thank the Reviewers for the reviews. I would like to say that I am immensely grateful to the Reviewers for all remarks.
All comments on the manuscript have been considered and improvements have been applied.
All changes are marked in the manuscript.
ANSWERS TO THE COMMENTS OF REVIEWER
Comment 1
Just explain why o pH 5 in the Preparation of Solutions and Sorption Experiment, and add the range of solutions of ICP.
Answer:
Thank you very much for your review. The pH 5 was chosen for the study, as this is the value below which ion precipitation occurs (Dobrowolski et al., 2017 [24]).
The concentration of the solutions used in ICP have been added.
“CCS-6 obtained from Inorganic Ventures (USA) was used as a standard solution for determination of elements (100µg/ml in 7% HNO3, Inorganic Ventures, USA).”

Round 2
Reviewer 1 Report
The manuscript can be accepted for publication in the present form.
Reviewer 2 Report
Thanks to the authors for their kind response; however, the majority of main concerns in the papers are not resolved, like the characterization and studying different factors which could affect the %R.